# Are Internet Information Sources Helpful for Adult Crohn’s Disease Patients Regarding Nutritional Advice?

**DOI:** 10.3390/jcm13102834

**Published:** 2024-05-11

**Authors:** Stefano Fusco, Katharina Briese, Ronald Keller, Carmen T. Schablitzki, Lisa Sinnigen, Karsten Büringer, Nisar P. Malek, Eduard F. Stange, Thomas Klag

**Affiliations:** 1Department of Gastroenterology, Gastrointestinal Oncology, Hepatology, Infectious Diseases and Geriatrics, University Hospital Tuebingen, Otfried-Müller-Strasse 10, 72076 Tuebingen, Germany; carmen.schablitzki@med.uni-tuebingen.de (C.T.S.); lisa.sinnigen@med.uni-tuebingen.de (L.S.); karsten.bueringer@med.uni-tuebingen.de (K.B.); nisar.malek@med.uni-tuebingen.de (N.P.M.); eduard.stange@icloud.com (E.F.S.); 2Clinic for Anaesthesiology, Operative Intensive Care, Emergency Medicine and Pain Therapy, Klinikum Stuttgart, Kriegsbergstr. 60, 70174 Stuttgart, Germany; k.briese@klinikum-stuttgart.de; 3Max-Planck-Institut für Biologie Tuebingen, Department Microbiome Science, Max-Planck-Ring 5, 72076 Tuebingen, Germany; ronald.keller@tuebingen.mpg.de; 4Bauchraum, Gastroenterologisches Zentrum, Bessemerstraße 7, 70435 Stuttgart, Germany; praxis@bauchraum-stuttgart.de

**Keywords:** Crohn’s disease, internet, nutrition, dietary counselling, IBD

## Abstract

**Background:** Adult patients suffering from Crohn’s disease (CD) are often dissatisfied with the information they receive from their physicians about nutrition and its impact on CD inflammation activity. Only a few publications are available about patients’ internet research on nutrition in CD. The study aim is to elucidate the internet information sources of adult CD patients regarding nutritional advice via a questionnaire. **Methods:** A questionnaire with 28 (general and specific) questions for outpatients at our tertiary center with CD was created and used for an analysis of their information sources about nutrition in CD. Four CD and/or nutritional medicine experts examined the 21 most relevant websites referring to nutritional advice for CD patients. **Results:** One hundred and fifty CD patients reported their Internet research behavior for nutritional advice and their dietary habits. Many CD patients prefer to consult the Internet instead of asking their general practitioner (GP) for nutritional recommendations. Most of the websites providing nutritional advice for CD patients are of very poor quality and cannot be recommended. We found significant correlations between (a) nutritional habits of CD patients, (b) their information sources and several demographic or CD-related factors. There is a lack of websites which provide high-quality, good nutritional advice to CD patients. **Conclusions:** The majority of the examined websites did not provide sufficient information according to the CD guidelines and nutritional medicine guidelines. A higher quality level of website content (e.g., on social media or on university/center websites) provided by experienced physicians is required to secure trustworthy and reliable nutritional information in CD.

## 1. Introduction

Crohn’s disease (CD) is a multifactorial, chronic inflammatory bowel disease, still with a poorly understood etiology that can also affect several extraintestinal organs and tissues [1,2,3,4]. There are different phenotypes of CD like the penetrating (fistulae) or the stenosing phenotype. Some CD patients develop a stenosis of the gastrointestinal tract or even fistulae (most frequently perianal fistulae) [5,6,7]. It has been hypothesized that CD may develop in persons with a genetic predisposition who experience defined environmental factors, an altered gut microbiome and a dysregulated immune response [8]. There are certain known risk factors (e.g., smoking, urban living, appendectomy, antibiotics) that may increase the risk of developing CD, but protective factors (like breastfeeding or rural living) are known [2]. Patients with inflammatory bowel disease (IBD) are at an increased risk of poor nutrition and malnutrition, particularly patients with active inflammation and therapy-refractory disease [9]. An increasing trend in the incidence and prevalence of IBD is depicted by genetics, gut microbes, as well as dietary and environmental influences as the major contributing factors. Among these, dietary factors in early life events such as maternal diet, antibiotics and formula feeding play a significant role in the development of gut microbiota. Gut microbiota dysbiosis is found to be closely associated with CD [10]. These have been considered to be Western diseases. Incidence and prevalence are increasing globally, particularly in industrialized and industrializing regions of the world such as Asia, the Middle East and Latin America [11,12]. In order to explain these increasing incidence rates, the role of diet has been closely examined. High dietary intake of total fats, polyunsaturated fatty acids (PUFAs), omega-6 fatty acids, red meat and sugar-sweetened beverages has been associated with an increased risk of CD in observational studies [13,14].

There is often a multidisciplinary approach to treating patients with intra- and extraintestinal manifestations (EIMs) of CD [15]. On the one hand, gastroenterologists as well as other experienced inflammation physicians prescribe immunosuppressant pharmakons like glucocorticoids, azathioprine, methotrexate or biologicals (e.g., integrin inhibitors (vedolizumab), TNF-alpha inhibitors (e.g., adalimumab, infliximab), interleukin inhibitors (risankizumab, ustekinumab)) or even small molecules (JAK inhibitors (upadacitinib)) for the treatment of CD and its EIMs [16]. On the other hand, a significant fraction of these treated patients are refractory to those pharmacological therapies, and they require a change of lifestyle, eating habits, physical exercises and other behavioral habits (particularly stopping smoking) [17,18,19]. They therefore need supplementary treatment including dietary counselling and dietary change [20]. Due to the inflammatory burden of the gastrointestinal tract, typical signs and symptoms of malnutrition can be detected; this can be iron deficiency anemia as well as a lack of water- and fat-soluble vitamins like vitamin D3 that can lead to osteopenia or even to osteoporosis [21,22]. One of the most common EIMs in CD is iron deficiency anemia that can lead to fatigue, headache, weariness and lack of concentration. This can be caused either by malnutrition or malassimilation due to intestinal malabsorption based on inflamed and dys-/hypotrophic small bowel villi. Certain nutrients (e.g., FODMAPs, gluten-based ingredients, red meat) can exacerbate the CD inflammation burden and should be avoided during CD inflammation activity [23,24]. Other foods (like chicken, fish, eggs, nuts, curcumin, vegetables and most fruits) are known to improve CD symptoms and can be eaten comfortably and even have an anti-inflammatory effect (e.g., curcumin) [24,25].

Several dietary strategies have been tested in CD patients during recent years. These include a CD exclusion diet (CDED), low fermentable oligosaccharides, disaccharides, monosaccharides and polyols (FODMAP), partial enteral nutrition (PEN), a specific carbohydrate diet (SCD) and exclusive enteral nutrition (EEN) which is recommended in pediatric patients [26,27,28,29].

CD patients often are interested in alternative methods to improve the CD symptoms. Making a dietary change or supplementary oral intake of phytopharmakons, dietary supplements or even homeopathic drugs can improve symptoms. A typical informational source for alternative or dietary treatment is the Internet and CD patients often receive their dietary advice from non-professional websites with occasional inappropriate recommendations.

The most common websites providing nutritional advice for CD patients are inadequate as the advice that is given is often inappropriate. There are only a few studies evaluating the role of Internet information sources on dietary counselling and dietary change in patients suffering from active CD [30,31].

In this study we elucidate the need for professional dietary counselling to better detect and improve malnutrition symptoms and the inflammatory activity of CD patients via nutritional advice.

## 2. Materials and Methods

This study was an observational prospective trial of adult CD patients who were seen and treated between January 2022 and April 2022 at the University Hospital Tuebingen, a tertiary center for IBD patients. The study protocol was approved by the Ethics Committee of the University Hospital Tuebingen (1232018BO2, approved 26 February 2018) and all patients signed an informed consent form before enrolment.

As part of the study, a printed questionnaire was created, which was then given to the CD cohort to complete. The survey took place in the Gastroenterological Outpatient Clinic of the Medical Clinic of the University Hospital Tuebingen. The questionnaire was distributed to patients at the IBD consultation. All patients underwent a physical examination, a bowel ultrasound, a blood sampling and a stool sampling as well as professional nutritional counselling by our nutrition support team. Participation was voluntary and anonymous. The questionnaire was handed out to patients by outpatient clinic staff when they registered. They had the opportunity to complete the form in the waiting room and then hand it back at the registration desk. The patient population thus comprised adult patients of all ages, the only inclusion criterion being a diagnosis of CD. The questionnaires were distributed between January and April 2022. One hundred and fifty questionnaires were collected in total. The questionnaire consists of twenty-seven questions including five about demographic characteristics, twelve about disease characteristics and ten questions about information sources and nutritional characteristics. The questions asked in the questionnaire are shown in Table 1 in a short overview. Twenty-six questions were closed-ended, the only open-ended question was the one about medication. Some items are subdivided into several subitems, such as the source of information (Internet, print media, general practitioner (GP), gastroenterologist (GE), patient information events or others) or alternative therapies like homeopathy, vitamin supplements, phytopharmakons, traditional Chinese medicine (TCM), physiotherapy, psychotherapy, lactose-free or gluten-free diet or the use of frankincense. Age category consists of three age groups (18–29 years, 30–49 years and 50 years or older). The smoker status was subclassified as never smoker, former smoker and current smoker. The BMI was calculated by us.

The “General Information” section records personal details such as age, height, weight, sex and duration of illness. The subjects’ smoking status is also recorded. The “Special Information” section deals with questions about the subjects’ CD. Question 8 is aimed at the intake of medication. The individual medications were asked about in the only open question. This is followed by a question on regular visits to the doctor with more detailed questions on the frequency of inpatient treatment, occurrence of deficiency symptoms, operations directly related to CD and, if indicated, information on the frequency of operations. The next question deals with the occurrence of extraintestinal symptoms. The questionnaire then moved away from the pure topic of CD to questions about diet and Internet use. Firstly, it was recorded whether the patients had ever received nutritional counselling and, furthermore, whether they use alternative therapies, with follow-up questions on consultation with treating physicians and knowledge of side effects. The test subjects were asked about the frequency of Internet use. Then, they were queried about the sources of information used when searching for information concerning health. This was followed by a question asking if the patients ever used the Internet to research their illness. Thereafter, a question asking the test subjects whether they had ever used the Internet to search for information about nutrition for CD was presented. This is specified in four additional questions: have tips from the websites been tried out, has an improvement or deterioration been noticed and did the test subjects inform their treating doctor about this. The last three questions deal with the topic of nutrition in CD—assessment of their own level of knowledge about nutrition in CD, the possible desire for further information on this topic and interest in professional nutritional counselling.

The second part of this study involved an expert analysis of the 21 websites that ranked highly in Google searches. This expert analysis included the compliance with guideline recommendations as well as analyzing whether the websites fulfilled formal and general criteria as mentioned in Section 3.8. Liebl et al. established an analysis system for websites (Appendix A) [32]. This analysis system was then published in *Oncology Research and Treatment* on 20 April 2016 by Herth et al. in the article “Internet Information for Patients on Cancer Diets—an Analysis of German Websites” [33]. Based on the analysis system in the above-mentioned articles, 17 general criteria and 6 formal criteria were defined. The analysis system is based on the Health on the Net Foundation (HONcode). There was a standardized scheme for the structure of the questions and possible answers. There were three possible answers, one of which had to be selected for each question. These were single-choice questions. The possible answers were: 1. the website fully agrees with the content of the criterion. 2. The website partially fulfills the content of the criterion, but there are some shortcomings. 3. The website contradicts the criterion in decisive statements or does not agree with it. The questions are structured as a Likert scale. Answer 1 is rated with one (1) point, answer 2 with three (3) points and answer 3 with five (5) points.

## 3. Results

### 3.1. Study Population Characteristics

The study recruited 150 patients (mean age 40.0 ± 15.5 years, 50.7% (*n* = 76) female, mean BMI 23.6 ± 4.7) (Table 2). Of them, 16.8% are current smokers. The mean CD duration is 15.3 years and about half (50.7%) of the study population experienced extraintestinal manifestation of CD. Of the patients, 56% are still biological therapy naïve and 78% of the study population suffer/suffered from malnutrition due to past or current CD activity.

The age of the study cohort was divided into three age groups, ranging from 18 to 29 years in the youngest age group, from 30 to 49 years in the middle group and all patients of at least 50 years in the third age group. Figure 1 illustrates the fractions of the different age groups, which have similar proportions.

The distribution of the demographic characteristics like sex and smoker status across the different age groups is illustrated in Figure 1 and Figure 2. While sex does not differ significantly between the age groups, there is an increased rate of former smokers and reciprocally a decreased rate of never smokers with age. The fraction of current smokers is fairly stable at between 12 and 19% in the different age groups. We divided the cohort into three age subgroups, as it is well known from the literature that younger and elderly patients differ in their behavior and some disease characteristics like BMI, smoker status or even the number of CD-associated operations.

The body mass index (BMI) of the whole study cohort has a mean value of 23.6 kg/m^2^ over all age groups, but there are differences between the age groups. Figure 3 presents the increase in BMI from the youngest to the oldest age group. Only the cohort aged 50 or more had a mean BMI of 25.8 kg/m^2^, which corresponds to being overweight.

### 3.2. Sources of Information for General Health Issues

Our patient cohort reported their preferred sources when searching for health information. The results are demonstrated in Table 3. Specialists, particularly in gastroenterology, are the source most frequently mentioned when patients inquire about health issues with a rate of 72.8%. The second preferred source was the Internet with 71.3% of the given answers. General practitioners were only named in third place as a source of information with 61.8%. The remaining sources each accounted for between 4% and 11% of the answers given. Print media such as brochures or books are used less frequently and account for only 4–8% of sources.

### 3.3. Alternative Therapy Modalities or Substances

Many CD patients reported the intake of alternative substances like frankincense or homeopathic agents and special exclusion diets such as lactose-free or gluten-free diets or vitamin supplements (see Table 4). Other patients of the study undergo psychotherapy, physiotherapy, traditional Chinese medicine (TCM) or mindfulness training or meditation. Vitamin supplementation is the most frequent alternative therapy modality with a fraction of 37.8%.

### 3.4. Factors Associated with Different Behavioral Items

Table 5 shows the significant univariate regression analysis accompanied by the multivariate regression analysis regarding different behavioral items. We only specify and list significant correlations in Table 5. Each item in bold represents the reference group for the following items/factors in the box underneath. All items/factors of the same box were analyzed together in a multivariate regression analysis. Patients who underwent nutritional counselling were significantly more likely to make regular visits to the doctor and took part at patient information events. They also reported significantly more vitamin supplementation than patients without the experience of nutritional counselling. In contrast, the same study population fraction used the Internet less frequently as a source of information. Patients with malnutrition or a history of malnutrition showed a significantly increased risk of requiring biological therapy. Furthermore, they had a higher number of operations. In contrast, these patients were less likely to experience symptom improvement after following dietary advice.

The study population that received regular medical consultation (defined as an outpatient medical consultation every three to twelve months) were more likely to have experienced a biological treatment as well as nutritional counselling. On the other hand, those patients showed a poorer experience with a lactose-free diet or the intake of frankincense. The same cohort received significantly more specialist information including nutritional advice from gastroenterologists.

While all univariate analyses in Table 5 reached a level of significance, the multivariate analysis only attained significance for the following factors:Combination of nutritional counselling and vitamin supplementation.Combination of malnutrition and CD duration as well as experience of biologicals and a reciprocal relationship regarding malnutrition and improvement of CD symptoms after nutritional advice.Reciprocal relationship regarding the combination of regular medical consultation and lactose-free diet.

### 3.5. Factors Associated with Different Nutritional Items

Patients who tried a lactose-free diet were significantly more often former smokers. A surprisingly low number of never smokers tried a lactose-free diet, as shown in Table 6. The same cohort was less likely to require CD medication. They also had a reduced frequency of medical consultation compared to patients without a lactose-free diet. In contrast, the same cohort often took their health and CD-related information from printed media like books, brochures or booklets. If this study cohort tried other nutritional advice as well there was also a positive correlation between a lactose-free diet and an improvement of CD-associated symptoms. Conversely, this means that patients without a lactose-free diet neither benefited from an improvement in the symptoms of CD, when they followed dietary advice, nor have been willing to receive professional nutritional information from their GP. A lactose-free diet was more common in former smokers (with an OR of 4.03) while the majority of never smokers seem to avoid this specific diet (OR 0.13). Patients on a lactose-free diet had a low odds ratio for current medication requirements and they were less likely to have regular medical consultations, which means at least once per year.

Paradoxically, when patients followed a gluten-free diet, a significant number of them suffered a deterioration of their CD symptoms, when following other dietary advice. The gluten-free diet cohort had a higher rate of biological treatment than patients not avoiding gluten in their nutrition. In patients on a gluten-free diet, the number of CD-associated operations was significantly higher than in the other cohort. In contrast to the study population on a lactose-free diet, the symptoms of CD worsened in the cohort on a gluten-free diet, when dietary recommendations were followed (OR 0.41).

A balanced (unspecific) diet with vitamin supplementation is more often correlated with a higher rate of physiotherapy utilization and professional nutritional counselling. The group of patients who seek advice from a gastroenterologist on health issues relating to CD and on special nutrition are more likely to take vitamin supplements. This also applies to the use of professional nutritional counselling.

The intake of frankincense (a plant with antioxidant and anti-inflammatory effects from active ingredients in the genus Boswellia) is significantly correlated with a lower rate of patients requiring medication and with a lower probability of regular medical consultations. The cohort consuming frankincense is associated with a slightly higher rate of hospital admissions. Frankincense use has a high OR regarding the simultaneous intake of homeopathic agents.

Patients from our study who have an affinity for homeopathy are more likely to attend patient information events and are significantly more likely to perform meditation exercises and take frankincense. The probability of suffering from EIMs is less likely in this cohort using homeopathic agents.

The intake of food supplements often leads to the additional intake of phytopharmaceuticals. In this cohort, however, the willingness to follow nutritional advice is significantly lower. The BMI is also lower in this group compared to patients not taking dietary supplements. We only specify and list significant correlations in Table 6.

Table 6 points out the following independent nutritional or supplemental factors, analyzed with the multivariate logistic regression:Reciprocal correlation between lactose-free diet and never smoker status as well as receiving nutrition information from a GP.Correlation between lactose-free diet and patient nutrition information from printed media together with CD-associated symptom improvement after following dietary advice.A slightly longer CD duration is seen in the cohort on a gluten-free diet. The same cohort more often has a need for biological therapy but this is reciprocally correlated to CD symptom improvement after following nutritional advice.Patients on vitamin supplementation show a higher rate of receiving information from their GE.The intake of frankincense leads to a slightly higher rate of hospital admission.Patients with intake of homeopathic drugs have higher requirements for medication. They also participate more frequently in patient information events. On the other hand, they are less frequently affected by EIMs.

### 3.6. Factors Associated with Different General Health Information Items

Patients who use the Internet as source to inform themselves about health issues are more often younger than 50 and are significantly less likely to seek nutritional advice. They are more often interested in information about CD and nutritional advice. This cohort profits from a higher rate of CD symptom improvement following nutritional advice. None of these factors are independent as shown in the multivariate logistic regression analysis.

If print media was the preferred source of information for health topics, there was a significantly higher correlation with the adherence to lactose-free diet and to traditional Chinese medicine (TCM). The same group complies more frequently to given nutritional advice and is better informed about nutrition in CD.

The patient group that visits the GP to obtain information about health issues is often over 50 years old and is significantly less likely to follow a lactose-free diet. They are also less likely to follow nutritional advice.

In contrast to the group consulting their GP, the patients that ask their GE for information about health are significantly less likely to be younger than 30 years old and to be current smokers. They show a higher BMI, take vitamin supplements with a higher probability and usually attend regular medical consultations.

The patients that take part at patient information events exhibit a higher number of CD medications as well as a higher number of CD-associated operations. This cohort also uses homeopathic agents more often and goes to nutritional counselling.

The multivariate logistic regression analysis demonstrated significance for independent factors associated with different general health information items (see Table 7).

These factors are the combination of print media and lactose-free diet, TCM and information about nutrition in CD as well as the consultation of GEs combined with a higher BMI or a higher probability of oral vitamin supplementation. The last correlation of Table 8 is between attending patient information events and the intake of homeopathic drugs.

### 3.7. Factors Associated with Different Specific CD/Nutrition Information Items

The patients who use the Internet to obtain information about CD are very rarely over 50 years old and have, on average, a lower number of CD-associated operations. Both points are independent factors in the multivariate logistic regression analysis (see Table 8).

If the Internet was used to inform about CD associated nutritional advice, the cohort was more likely aged under 50 years and had significantly more often had a lower BMI. The majority of those patients were male. All factors were independent in the multivariate analysis.

The patients that tested the CD-specific nutritional advice showed a significantly higher rate of CD-associated operations and also used printed patient information as an added information source. The same cohort was less likely to attain the information from their GP. These three factors were also independent in the multivariate analysis.

On the one hand, patients that experienced an improvement of CD symptoms after following nutritional advice from the Internet were more willing to follow a lactose-free diet but had a lower rate of malnutrition. On the other hand, the cohort with a history of impairment of CD symptoms after following nutritional advice from Internet was more often associated with a gluten-free diet. They were more frequently never smokers. A gluten-free diet was detected as an independent factor for impairment of CD symptoms in the multivariate analysis.

### 3.8. Expert Analysis of the Websites regarding Compliance with Guideline Recommendations

Four experts (experienced gastroenterologists and nutritionists) analyzed the first 21 websites that appeared during Internet research via Google when searching for “Crohn’s disease and nutrition” on 15 August 2021. The experts checked whether the websites had followed the 17 recommendations of the German CD and nutrition guidelines. They also checked the websites with a standardized evaluation sheet, which contained several items like neutrality (no advertising), relevance of information, transparency, information on references and resources or focus on the patient. The analyzed websites were divided into four different categories: twelve commercial websites, three non-commercial websites, two booklets and four other websites.

The 21 websites have been ranked with a normalization of the values from 0 (best) to 100 (worst) regarding the compliance to guideline recommendations, which is shown in Figure 4. It can be seen that the three non-commercial websites obtained the best results. The commercial websites achieved a value of 76.6, the booklets a value of 80.9. The websites classified as “others” were ranked very low with a value of 84.9 points. Better values have been obtained for general and formal criteria of the websites (see Figure 5). The best mean value of 46.4 points was achieved by the online booklets. The commercial, the non-commercial and the other websites did not differ significantly with ranges from 51.2 to 54.2 points on a scale from 0 to 100, which corresponds to an average performance.

The compliance of CD guideline recommendations was better on all websites compared to compliance of nutrition guidelines, which was very bad, ranging from 83 to the worst value of 100 points.

None of the 21 websites scored well. None of the mentioned websites can be recommended for patient research as of 2023.

## 4. Discussion

In our observational questionnaire study, we investigated the behavior and needs of 150 CD patients regarding their sources of information when it comes to health issues related to CD or dietary advice in CD. We provided a questionnaire with 28 questions which was completed in our outpatient care unit at the tertiary university center in Tuebingen. One of the main concerns that emerged from this study was a lack of professional nutritional advice on 21 popular German websites that were surveyed. Most of these websites, found via Google search with the search term “Crohn’s disease nutrition”, are inappropriate regarding nutritional advice. The websites analyzed for this study presented a mismatch with national guidelines for nutrition in CD or the national guideline of CD. They were ranked by a standardized questionnaire with a normalized score between 0 (best) and 100 (worst). Non-commercial websites obtained the best score with 73.4 points regarding their compliance with guideline recommendations. The same websites scored better in terms of formal and general criteria with an average score around 50 points. This is congruent to other published studies. To date, there is a great lack of high-quality websites that provide clear information and guideline-based advice on nutrition in CD [31,34]. Nevertheless, there has been progress in online searches about IBD or CD. Since September 2022, a new Internet tool called ChatGPT appeared in the field of AI. However, the publication by Gravina et al. shows that ChatGPT is not yet able to replace a doctor in terms of information on CD and nutrition. Some of its information is outdated, does not provide any evidence or is simply wrong [35].

Gradually, video platforms such as YouTube, Instagram or TikTok are including videos on nutritional advice and daily management of CD. In those online videos, investigated by Gkikas et al., scientific evidence was cited in 2 (3%) of 76 patient videos compared with 25 (35%) of 71 qualified healthcare professional videos. Gkikas’ lab group have identified nutritional advice proposed as beneficial or detrimental in the management of IBD [36]. Amante et al. conducted a retrospective analysis with the aim to determine whether the difficulty to access healthcare services was associated with increased use of the Internet to obtain health information. They discovered that people experiencing problems accessing healthcare services are more likely to select using the Internet to obtain health information [37].

In our study, we found that patients use the Internet (71.3%) significantly more often than their GP (61.8%) as a source of information on health issues. Only specialists like gastroenterologists (72.8%) are more often asked for health information by the patients.

A US study showed that more than half of the population suffering from inflammatory bowel disease (IBD) use the Internet as a primary source of information on their condition [38]. Garg et al. analyzed top influencers on the topic of IBD on X (formerly Twitter) and correlated the relevance of their social media engagements with their professional expertise or academic productivity. Analyzing 100 X influencers on IBD topics, they have observed a positive correlation between the X topic score and the h-index of the influencers. Most of those influencers were gastroenterologists [38]. An Italian multicentric trial evaluated the use of the Internet amongst 495 Italian IBD patients. Approximately half of the patients in Italian IBD referral centers used the Internet to gather IBD-related information. This usage was positively correlated to disease activity and severity. A large majority of patients requested the IBD referral centers to have their own IBD-dedicated website [39]. A US cross-sectional study demonstrated that fifty percent of respondents (IBD patients) could not rate the quality of IBD information posted online, particularly on social media. Patients’ concern regarding the use of social media included privacy/confidentiality and lack of trust in the information posted [40].

Our study showed that there are several independent correlating factors regarding CD patients and their nutritional behavior or diets. We were also able to identify significant correlations between the preferred sources of information and specific characteristics associated with CD. One of the important results was the correlation between a lactose-free diet and the improvement of CD-associated symptoms, whereas a gluten-free diet impaired the CD activity. Furthermore, CD patients that underwent nutritional counselling have a higher probability of vitamin supplementation. In addition, patients who obtained information about nutrition on the Internet and who tested this nutritional advice were significantly less likely to talk to their GP about the same topics. This is consistent with some published studies [41,42]. Surprisingly, there was an association between never smokers and an impairment of CD symptoms when following nutritional advice. Study patients who asked their GE for health information about CD and nutrition were more likely to opt for vitamin supplementation intake. A positive correlation between the attendance at patient information events and the intake of homeopathic agents could be recognized. Also, attendance at patient information events correlated to (a) the amount of medications and (b) Crohn’s-specific operations. The number of medications and the Crohn’s-specific operations were not independent of each other. This may be due to the fact that patients with higher inflammatory activity, possibly with a stenosing or penetrating course, have a greater need for more intensive information which can be obtained at these special information events. If print media were the information source for health and CD topics, patients have an 11-fold higher probability to have a lactose-free diet and to use TCM as alternative treatment. The same cohort was more likely to show interest in information about nutrition with an odds ratio of more than 26.

It is known that CD patients try different diets, such as a lactose-free diet or gluten-free diet or a low-FODMAP diet to improve the CD activity or even to achieve clinical remission. Actually, the CD elimination diet (CDED) is the only evidence-based diet considered for adult and pediatric CD patients [28,43,44,45]. All other CD nutrition studies were too heterogeneous regarding the effects on CD activity. Hence, a lactose-free or gluten-free or low-FODMAP diet cannot be recommended due to a lack of evidence. Nevertheless, we found a correlation between patients that underwent a lactose-free diet and their smoker status. While never smokers had a negative correlation regarding a lactose-free diet, the former smokers were more likely to try this diet. A reciprocal association could be demonstrated between a lactose-free diet and the need for medication, regular medical consultation as well as consulting a GP for health information. So, it appears that patients on a lactose-free diet initiated it without the advice of their GP or any other physician, because they obtained most of their information from print media.

We found that patients reporting intake of frankincense had a five-fold higher probability to use homeopathic remedies as well. More frequently, those who take homeopathic remedies show a positive correlation with meditation. Finally, the intake of nutrition supplements is correlated with an affinity for phytopharmaceuticals. Certain phytopharmaceuticals like curcumin have shown a higher clinical and endoscopic remission rate in CD. However, frankincense led to contradictory study results, so it cannot be recommended for CD patients [46,47]. The desire to improve CD-related symptoms often leads to the use of alternative therapies and nutritional counselling. Gastroenterologists as well as GPs need a better understanding of patients’ needs regarding nutrition and achieving CD remission. Our study shows that there is a large gap between professional nutritional counselling and the available websites with nutrition advice. We encourage other IBD centers to instruct their CD patients in certain live situations with impairment of quality of life because of CD burden and to hand out a checklist. Objective tests like ultrasound and inflammation markers (e.g., calprotectin) should be carried out to check the patients’ compliance.

A limitation of our study is that it is performed as a monocentric trial and that the questions of the questionnaire were not able to analyze a correlation between good responders to certain therapy modalities and the degree of internet searching.

## 5. Conclusions

Patients often need professional nutritional counselling to cope with everyday life when suffering from CD inflammation activity and EIMs. An interdisciplinary team consisting of gastroenterologists, nutritionists, dieticians, pharmacologists and psychologists should constantly accompany patients with CD to prevent undernourishment and to recognize malnutrition during inflammatory episodes early as well as provide additional online support via social media. The most common websites providing nutritional advice for CD patients are inadequate. Social media influencers are starting to develop a crucial role as an information source for patients. About 70% of the patients obtain their information about CD and nutrition via the Internet. A higher quality level of website content (e.g., on social media or on university/center websites) provided by experienced physicians is required to secure trustworthy and reliable nutritional information on CD.

## Figures and Tables

**Figure 1 jcm-13-02834-f001:**
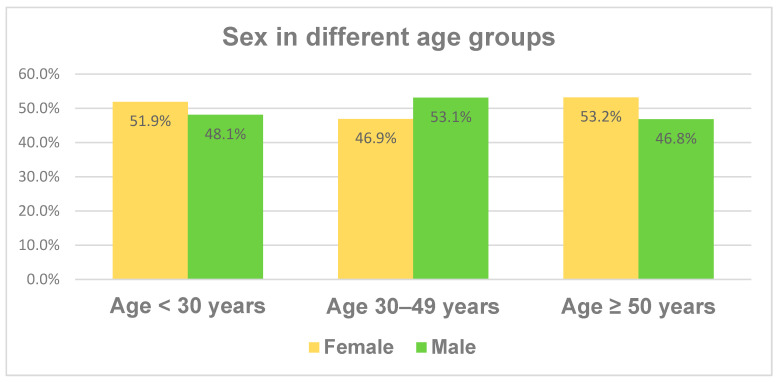
Demographic distribution depending on patient age.

**Figure 2 jcm-13-02834-f002:**
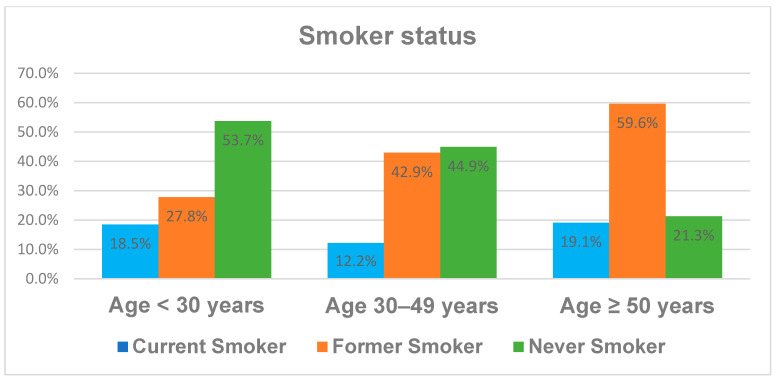
Smoker status depending on patient age.

**Figure 3 jcm-13-02834-f003:**
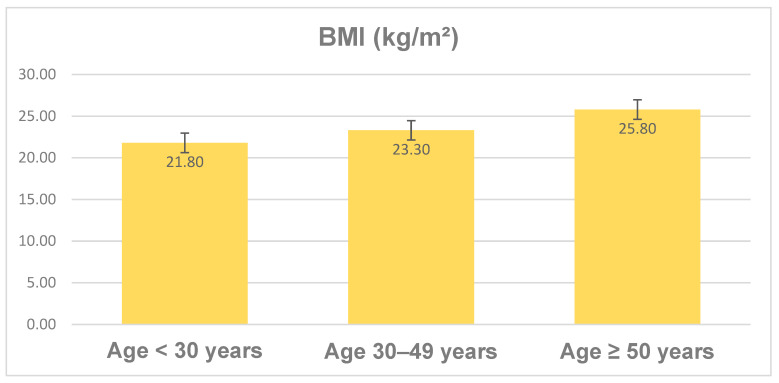
BMI distribution (means) depending on patient age (in years) with standard deviation.

**Figure 4 jcm-13-02834-f004:**
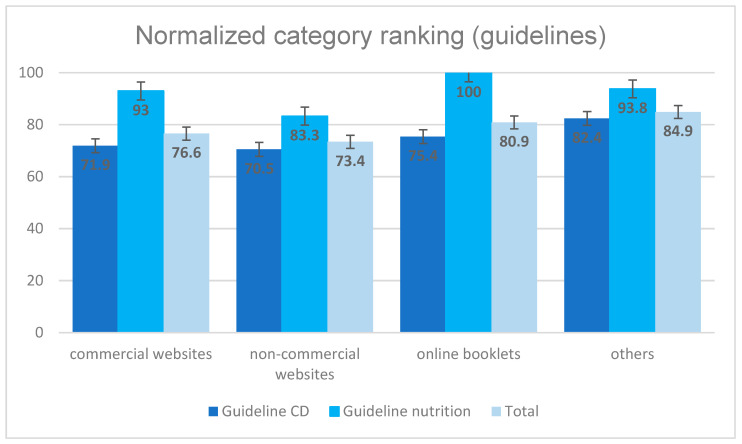
Website evaluation regarding guideline recommendations.

**Figure 5 jcm-13-02834-f005:**
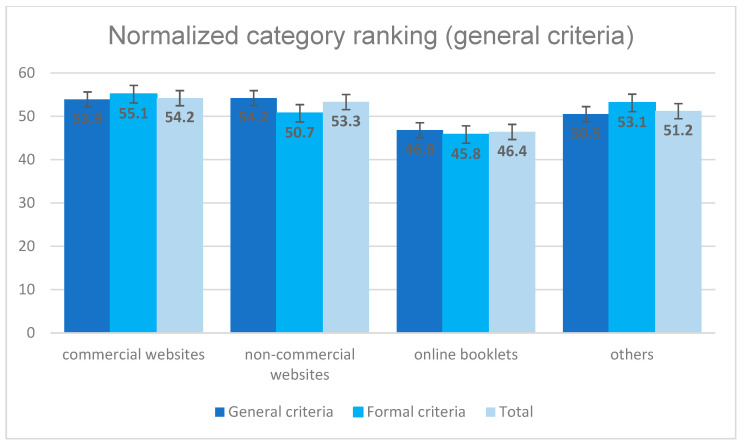
Website evaluation regarding formal and general criteria.

**Table 1 jcm-13-02834-t001:** CD patient’s questionnaire (27 questions).

What is your current age?
What is your current body height?
What is your current body weight?
What is your sex?
What is your current smoker status?
How many years ago was the diagnosis made?
What is your actual CD medication? (open ended question)
Have you ever experienced a biological therapy?
How many CD medications are you taking?
Have you ever had medication interactions?
Which medication interactions have you experienced?
Do you regularly visit the doctor/GP?
How often have you been hospitalized due to CD?
Have you ever suffered from malnutrition?
Have you ever had surgery for CD?
How many CD related operations have you had?
Do you have a history of EIM?
Have you ever had nutritional counselling?
Did you try alternative therapy modalities?
How often do you use the Internet?
What are your sources of health information?
Do you use the Internet as an information source on CD?
Do you use the Internet as an information source on nutritional advice?
Have you implemented the nutritional advice?
Have you experienced an improvement in CD symptoms after following the dietary advice?
Have you experienced a worsening in CD symptoms after following the dietary advice?
Do you feel the need for nutritional counselling?

**Table 2 jcm-13-02834-t002:** Demographic and clinical characteristics of the study population.

**Demographic Characteristics**	
Age (years, mean ± sd)	40.0 ± 15.5
Sex—female % (*n*)	50.7 (76)
Current smoking % (*n*)	16.8 (25)
BMI at baseline (kg/m^2^, mean ± sd)	23.6 ± 4.7
**CD Characteristics**	
Disease duration (years, mean ± sd)	15.3 ± 12.6
EIM % (*n*)	50.7 (74)
Biological therapy naïve % (*n*)	56.0 (84)
Experience of malnutrition % (*n*)	78.0 (117)

Abbreviations: BMI: body mass index; EIM: extraintestinal manifestation; sd: standard deviation.

**Table 3 jcm-13-02834-t003:** Ranking of sources to obtain health information.

Ranking Sources of Information		
Sources of Information	Amount	
	Absolute	%
Gastroenterologist (GE)	99	72.8%
Internet	97	71.3%
General Practitioner (GP)	84	61.8%
Miscellaneous	16	11.8%
Books	12	8.8%
Booklets	9	6.6%
Patient information events	7	5.1%
Brochures	6	4.4%

Patients could choose one or more items, so the sum is higher than 100%.

**Table 4 jcm-13-02834-t004:** Ranking of alternative therapy modalities/substances.

Ranking Alternative Therapy Modalities/Substances		
Alternative Therapy Modalities/Substances	Amount	
	Absolute	%
Vitamin supplementation	31	37.8%
Miscellaneous	19	23.2%
Lactose-free diet	17	20.7%
Phytopharmaceuticals	13	15.9%
Physiotherapy	13	15.9%
Homeopathic agents	12	14.6%
Meditation/Mindfulness training	11	13.4%
Psychotherapy	10	12.2%
Special herbal infusion	10	12.2%
High-caloric food supplements	9	11.0%
Frankincense	8	9.8%
Gluten-free diet	7	8.5%
TCM	6	7.3%
Prebiotics/Probiotics	5	6.1%

Patients could choose one or more items, so the sum is higher than 100%.

**Table 5 jcm-13-02834-t005:** Univariate and multivariate regression analyses of factors associated with different behavioral items.

	Univariate	*p* Value	Multivariate	*p* Value
**Nutritional counselling**				
CD duration	1.03 (1.01–1.06)	0.022	0.98 (0.94–1.02)	0.332
Regul. medical consultation	3.27 (1.03–10.37)	0.045	8.08 (0.90–72.25)	0.062
Internet	0.44 (0.21–0.92)	0.029	0.38 (0.11–1.34)	0.131
Patient information event	11.51 (1.40–94.56)	0.023	3.23 (0.32–32.31)	0.319
No. of operations	1.15 (1.02–1.29)	0.023	1.03 (0.87–1.22)	0.77
Vitamin supplementation	2.77 (1.11–6.89)	0.028	**3.21 (1.15–8.97)**	**0.026**
**Malnutrition**				
CD duration	1.06 (1.02–1.11)	0.006	**1.06 (1.01–1.12)**	**0.033**
Biologicals	2.21 (1.01–4.86)	0.049	**3.69 (1.31–10.4)**	**0.013**
No. of operation	1.36 (1.05–1.76)	0.021	1.23 (0.92–1.66)	0.165
Improvement after nutritional advice	0.41 (0.17–0.99)	0.049	**0.34 (0.12–0.97)**	**0.044**
**Regul. Medical consultation**				
Medication	4.57 (1.62–12.89)	0.004	1.51 (0.18–12.46)	0.704
Biologicals	6.38 (2.00–20.37)	0.002	7.18 (0.91–56.69)	0.062
Nutritional counselling	3.27 (1.03–10.37)	0.045	3.79 (0.58–24.55)	0.163
Lactose-free diet	0.23 (0.06–0.87)	0.030	**0.15 (0.02–0.94)**	**0.043**
Frankincense	0.18 (0.04–0.87).	0.034	0.27 (0.03–2.36)	0.239
Information from GE	4.62 (1.70–12.57)	0.003	2.93 (0.57–14.96)	0.806

The headings in bold are reference groups for the following variables.

**Table 6 jcm-13-02834-t006:** Univariate and multivariate regression analyses of factors associated with different nutritional and supplemental items.

	Univariate	*p* Value	Multivariate	*p* Value
**Lactose-free diet**				
Former smoker	4.03 (1.30–12.52)	0.016	0.31 (0.48–2.06)	0.227
Never smoker	0.13 (0.03–0.58)	0.008	**0.02 (0.0–0.28)**	**0.004**
Medication	0.27 (0.08–0.98)	0.048	0.75 (0.13–4.30)	0.746
Regul. medical consultation	0.23 (0.06–0.87)	0.030	0.24 (0.03–1.94)	0.181
Printed patient information	3.63 (1.24–10.64)	0.019	**6.7 (1.34–33.59)**	**0.021**
Information from GP	0.32 (1.11–0.94)	0.038	**0.16 (0.04–0.75)**	**0.020**
Improvement after nutritional advice	2.96 (1.01–8.66)	0.047	**4.81 (1.13–20.48)**	**0.034**
**Gluten-free diet**				
CD duration	1.06 (1.02–1.11)	0.006	**1.06 (1.01–1.12)**	**0.033**
Biologicals	2.21 (1.01–4.86)	0.049	**3.69 (1.31–10.4)**	**0.013**
No. of CD operations	1.36 (1.05–1.76)	0.021	1.23 (0.92–1.66)	0.165
Improvement after nutritional advice	0.41 (0.17–0.99)	0.049	**0.34 (0.12–0.97)**	**0.044**
**Vitamin supplementation**				
No. of medications	1.53 (1.05–2.23)	0.024	1.34 (0.90–2.02)	0.152
Nutritional counselling	2.77 (1.11–6.89)	0.029	2.57 (0.96–6.88)	0.060
Physiotherapy	4.41 (1.18–16.52)	0.026	2.68 (0.64–11.15)	0.176
Information from GE	6.67 (1.44–30.79)	0.015	**5.59 (1.14–27.30)**	**0.034**
**Frankincense**				
Medication	0.20 (0.04–0.98)	0.047	0.16 (0.19–1.43)	0.101
Regul. medical consultation	0.18 (0.04–0.87)	0.034	0.26 (0.04–1.98)	0.195
Inpatient care	1.19 (1.04–1.35)	0.008	**1.22 (1.04–1.43)**	**0.016**
Homeopathy	5.00 (1.02–24.51)	0.047	6.16 (0.76–50.08)	0.089
**Homeopathy**				
EIMs	0.45 (0.21–0.98)	0.045	**0.42 (0.18–0.98)**	**0.047**
Frankincense	5.00 (1.02–24.51)	0.047	3.88 (0.62–24.46)	0.149
Meditation	5.24 (1.25–21.77)	0.024	**7.38 (1.36–40.10)**	**0.021**
Patient information event	6.33 (1.22–32.94)	0.028	**14.15 (2.06–97.02)**	**0.007**
**Nutritional supplement**				
BMI	0.73 (0.56–0.96)	0.022	**0.75 (0.56–0.99)**	**0.049**
Phytopharmaceuticals	5.33 (1.36–20.97)	0.017	**5.43 (1.21–24.43)**	**0.027**
Interest in nutritional advice	0.23 (0.06–0.89)	0.034	0.32 (0.07–1.41)	0.131

The headings in bold are reference groups for the following variables.

**Table 7 jcm-13-02834-t007:** Univariate and multivariate regression analyses of factors associated with different general health information items.

	Univariate	*p* Value	Multivariate	*p* Value
**Internet**				
Age (>50 a)	0.20 (0.09–0.43)	0.001	0.31 (0.08–1.15)	0.081
CD duration	0.96 (0.93–0.98)	0.005	1.00 (0.96–1.05)	0.822
Nutritional counselling	0.44 (0.21–0.92)	0.029	0.37 (0.13–1.08)	0.070
Information about CD	16.05 (3.29–78.30)	0.001	3.91 (0.30–51.77)	0.301
Information about nutrition	3.28 (1.48–7.28)	0.004	2.06 (0.57–7.63)	0.278
Improvement after nutritional advice	4.34 (1.21–15.53)	0.024	2.58 (0.66–10.12)	0.175
**Print media**				
Lactose-free diet	3.63 (1.24–10.64)	0.019	**11.73 (2.05–67.04)**	**0.006**
TCM	6.19 (1.06–36.24)	0.043	**11.80 (1.18–118.42)**	**0.036**
Tested nutritional advice	8.48 (1.08–66.63)	0.042	3.16 (0.24–42.53)	0.385
Information about nutrition	5.62 (1.62–19.53)	0.007	**26.25 (2.39–287.99)**	**0.008**
**GP**				
Age (>50 a)	3.26 (1.43–7.45)	0.005	2.42 (0.66–8.89)	0.184
Lactose-free diet	0.33 (0.11–0.94)	0.038	0.56 (0.16–1.88)	0.346
Tested nutritional advice	0.23 (0.06–0.85)	0.028	0.16 (0.02–1.44)	0.103
**GE**				
Age (<30 a)	0.36 (0.17–0.79)	0.010	0.38 (0.10–1.42)	0.383
BMI	1.20 (1.07–1.34)	0.002	**1.65 (1.21–2.25)**	**0.001**
Current smoker	0.38 (0.15–0.95)	0.039	1.88 (0.21–17.32)	0.576
Regul. medical consultation	4.62 (1.70–12.57)	0.003	2.50 (0.52–12.08)	0.255
Vitamin supplementation	6.67 (1.44–30.79)	0.015	**6.52 (1.05–40.37)**	**0.044**
**Patient information event**				
No. of medications	1.77 (1.09–2.86)	0.020	1.66 (0.85–3.24)	0.138
No. of CD operations	1.23 (1.05–1.44)	0.012	1.19 (0.96–1.48)	0.118
Nutritional counselling	11.51 (1.40–94.56)	0.023	6.04 (0.62–59.17)	0.123
Homeopathy	6.33 (1.22–32.94)	0.028	**12.66 (1.64–97.64)**	**0.015**

The headings in bold are reference groups for the following variables.

**Table 8 jcm-13-02834-t008:** Univariate and multivariate regression analyses of factors associated with different specific CD/nutrition information items.

	Univariate	*p* Value	Multivariate	*p* Value
**Internet CD**				
Age (>50 a)	0.15 (0.04–0.59)	0.007	**0.13 (0.02–0.94)**	**0.043**
CD duration	0.94 (0.89–0.98)	0.010	1.00 (0.94–1.07)	0.953
No. of CD operations	0.81 (0.70–0.94)	0.006	**0.81 (0.66–0.99)**	**0.039**
**Internet CD/nutritional advice**				
Age (>50 a)	0.26 (0.12–0.57)	0.001	**0.34 (0.15–0.78)**	**0.011**
BMI	0.91 (0.84–0.97)	0.011	**0.92 (0.85–0.99)**	**0.049**
Sex	0.41 (0.19–0.88)	0.022	**0.38 (0.17–0.86)**	**0.021**
**Tested Internet CD/nutritional advice**				
Operation	3.75 (1.32–10.63)	0.013	**4.10 (1.13–14.92)**	**0.032**
Printed patient information	8.48 (1.08–66.63)	0.042	**9.32 (1.05–83.23)**	**0.046**
Information from GP	0.23 (0.06–0.85)	0.028	**0.14 (0.03–0.74)**	**0.021**
**Improvement after Internet CD/nutrition advice**				
Malnutrition	0.41 (0.17–0.99)	0.049	0.36 (0.12–1.14)	0.083
Lactose-free diet	2.96 (1.01–8.66)	0.047	2.78 (0.93–8.32)	0.068
**Impairment after Internet CD/nutrition advice**				
Never smoker	10.50 (1.25–88.33)	0.030	9.58 (0.87–83.54)	0.283
Gluten-free diet	20.29 (1.63–252.86)	0.019	**19.33 (1.33–281.60)**	**0.030**

The headings in bold are reference groups for the following variables.

## Data Availability

The datasets employed and examined in this study can be obtained by contacting the corresponding author.

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
