# Peer review of "Are Internet Information Sources Helpful for Adult Crohn’s Disease Patients Regarding Nutritional Advice?"

_jcm, 2024, doi:10.3390/jcm13102834_

Round 1
Reviewer 1 Report
Comments and Suggestions for Authors
The work should be improved, presentation of results, tables, Materials and Methods section, as well as Conclusions. Detailed information is included below.
41. abbreviation IBD – no explanation in the text
192. abbreviation TCM – no explanation in the text, when it appears for the first time, and the explanation of the abbreviation is later in the article
There is no information in the title, abstract and chapter Material and Methods: that only adults were studied
Subsection 3.3 should be in the Materials and Methods section
209. Why was the BMI of patients asked when height and weight are above – can patients always calculate their BMI for themselves?
209 – Were all questions open? Characterize the questionnaire in more detail, which questions were open-ended? Maybe add this questionnaire as additional material
119. You wrote: Question 1 is aimed at the intake of medication" – whereas in Table 4 question 1 is different: What is your current age?
Table 5 is incomprehensible – is part of the correlation shown here? If so, you should add why you see this part and not the others. For example, what was the correlation between people browsing the internet and those who were malnourished? Similarly in other tables.
Section 3.8 line 370 – no information about it in Material and Methods – that there is a second/other part of the study consisting in checking the web pages for content and content
380-382. There it was written: The 21 websites have been ranked with a normalization of the values from 0 (best) to 100(worst) regarding the compliance to guideline recommendations, which is shown in figure 4. – no information on how points were awarded
385. What do general and formal criteria mean? How were the points awarded? There is no information about the results in the Results section. There are a lot of descriptions, but no data, maybe you need to add the number of people in the tables, not just the correlation
506-507. Since this sentence is in Conclusion: Social media influencers are starting to develop a crucial role as an information source for patients. – Where can I find support for this sentence in the results?
507-508 the same, where in the results I find support for these words: Women are less likely than men to talk to their physicians about Internet research and nutritional advice.
Please change the Conclusions – here should be the most important things from your study – specific 3-6 sentences of the most important from your study
543. In Acknowledgments: We do not thank the people if they are writer in this article – please see E. Stange
Reviewer 2 Report
Comments and Suggestions for Authors
Review for the manuscript
Are Internet information sources helpful for Crohn's disease patients regarding nutritional advice?
OVERALL COMMENTS
In this manuscript, the authors intended to investigate the behaviour of patients regarding Internet research for nu tritional advice and their dietary habits. The main findings include that “There is a lack of websites which provide high-quality, good nutritional advice to CD patients. The majority of the examined websites did not provide sufficient information according to the CD guidelines and nutritional medicine guidelines. A higher quality level of website content…provided by experienced physicians is required to secure trustworthy and reliable nutritional information in CD”
TITLE
It is adequate.
ABSTRACT
I miss a more structured summary showing the objectives of the study. Also, in this session, what does GP mean?
Why is the last paragraph separated from the rest of the text.
KEYWORDS
I suggest removing "questionnarire" from the Key-words.
INTRODUCTION
This section was well-performed. However, there is a need to include references published in 2023 and in 2024, mainly regarding the definitions of Crohn´s disease.
METHODS
This section is adequate and the authors mention the ethical principles that were used to perform the study.
RESULTS
This section is adequate. However, I suggest improving the figures. There must be standardization in the size of the graphics and the font size of the captions and other words inserted in each of them.
DISCUSSION
This section is complete. However, I suggest including references published in 2023 and 2024.
Minor: sometimes we see Crohn's disease (linke in line 502) and in many other times we see CD. Please check all over the text.
REFERENCES
The same comments I performed in the Introduciton and Discussion sections.
Comments on the Quality of English Language
Minor revision.
Reviewer 3 Report
Comments and Suggestions for Authors
Dear authors,
This is an interesting paper on the internet sources your Chron’s disease patients use for nutritional advice.
I have nothing to comment on the questionnaire and the analysis of answers provided and the correlation with the related bibliography.
However, my question is what consultation do you provide them? Do you have a nutritional counseling team? Do your patients clearly know the dos and don’ts for their lives? Do they follow them? Is there a correlation between well responders to whatever therapy and the degree of internet searching?
I would like please answers to my questions both in the material and methods as well as in the discussion.
Round 2
Reviewer 1 Report
Comments and Suggestions for Authors
In response to the replies sent to my comments, I am asking for clarification on some points.
1. Comments 5: Line 209: Why was the BMI of patients asked when height and weight are above – can patients always calculate their BMI for themselves? Response 5: Thank you again or your interesting advice. The BMI was calculated by us, not by the patients. We mentioned the calculation of the BMI by the study team in line 209.
If I understand correctly and you calculated your BMI yourself, why was this question included in the questionnaire? If this question was not included in the survey, why is it listed in Table 1? If you remove the BMI question and add a note saying that you calculated your BMI yourself, the number of questions will change and this will need to be corrected in the text, unless otherwise, please explain. Maybe, as I suggested, the entire questionnaire with all the questions should be included in the attachments as additional material.
2. Comments 7: Line 119: You wrote: Question 1 is aimed at the intake of medication" – whereas in Table 4 question 1 is different: What is your current age? Response 7: Thank you for mentioning thís issue. This was a mistake.We changed “Question 1” to “Question 8” in the text.
Question no. 1 is still written on line 131 - maybe divide the table into the parts you are writing about - "General Information" and "Special Infoarmtion" sections - then there will be no confusion
3. Comments 8: Table 5 is incomprehensible – is part of the correlation shown here? If so, you should add why you see this part and not the others. For example, what was the correlation between people browsing the internet and those who were malnourished? Similarly in other tables. Response 8: Thank you for your important note. Table 5 shows only the significant correlations between the variable in bold (“subtitles”) and the different variables listed below. For reasons of clarity, no reference was made to the numerous non-significant correlations.
Is the information "that not all data was included in table 5" included in the text? Maybe you can add them to the attachments as additional material.
Reviewer 3 Report
Comments and Suggestions for Authors
I am OK with your response - thank you!
Author Response
Dear reviewer,
we appreciate your time and klowledge und would like to thank you for your time and your review!